# Can fingernail quality predict bone damage in Type 2 diabetes mellitus? a pilot study

Praveer Sihota[1], Rimesh Pal[2], Ram Naresh Yadav[1], Deepak Neradi[3], Shailesh Karn[3], Vijay G. Goni[3], Siddhartha Sharma[3], Vishwajeet Mehandia[1], Sanjay Kumar Bhadada[2]*, Navin Kumar[1]*, Sudhaker D. Rao[4]

**1** Department of Mechanical Engineering, Indian Institute of Technology Ropar, Rupnagar, Punjab, India, **2** Department of Endocrinology, Post Graduate Institute of Medical Education and Research, Chandigarh, India, **3** Department of Orthopedics, Post Graduate Institute of Medical Education and Research, Chandigarh, India, **4** Department of Bone and Mineral Metabolism, Henry Ford Hospital, Detroit, MI, United States of America

* bhadadask@rediffmail.com (SKB); nkumar@iitrpr.ac.in (NK)

## Abstract

Type 2 diabetes mellitus (T2DM) adversely affects the normal functioning, intrinsic material properties, and structural integrity of many tissues, including bone. It is well known that the clinical utility of areal bone mineral density (aBMD) is limited to assess bone strength in individuals with T2DM. Therefore, there is a need to explore new diagnostic techniques that can better assist and improve the accuracy of assessment of bone tissue quality. The present study investigated the link between bone and fingernail material/compositional properties in type 2 diabetes mellitus (T2DM). For that, femoral head and fingernail samples were obtained from twenty-five adult female patients (with/without T2DM) with fragility femoral neck fractures undergoing hemi/total hip arthroplasty. Cylindrical cores of trabecular bone were subjected to micro-CT, and lower bone volume fraction was observed in the diabetic group than the non-diabetic group due to fewer and thinner trabeculae in individuals with T2DM. The material and compositional properties of bone/fingernail were estimated using nanoindentation and Fourier Transform Infrared Spectroscopy, respectively. Both bone/fingernails in T2DM had lower reduced modulus ($E_r$), hardness (H), lower Amide I and Amide II area ratio (protein content), higher sugar-to-matrix ratio, and relatively high carboxymethyl-lysine (CML) content compared with non-diabetic patients. Sugar-to-matrix ratio and relative CML content were strongly and positively correlated with $HbA_{1c}$ for both bone/fingernail. There was a positive correlation between bone and fingernail glycation content. Our findings provide evidence that the degradation pattern of bone and fingernail properties go hand-in-hand in individuals with T2DM. Hence, the fingernail compositional/material properties might serve as a non-invasive surrogate marker of bone quality in T2DM; however, further large-scale studies need to be undertaken.

**Data Availability Statement:** All relevant data are within the paper.

**Funding:** The IMP/2019/000150 Impacting Research, Innovation, and Technology (IMPRINT)-Ministry of Human Resource Development (IN)

https://imprint-india.org/, CRG/2018/002219
Department of Science & Technology, (IN) https://
dst.gov.in, and MoE/STARS-1/632 Ministry of
Education, Scheme for Transformational and
Advanced Research in Sciences (IN). The funders
had no role in study design, data collection, and
analysis, decision to publish, or preparation of the
manuscript.

**Competing interests:** The authors have declared
that no competing interests exist.

**Abbreviations:** T2DM, Type 2 diabetes mellitus;
NA, Not applicable; AGE, Advanced glycation end-
product; BV/TV, Bone volume/Trabecular volume;
Tb.Th, Trabecular thickness; Tb.N, Trabecular
number; Er, Reduced modulus; H, Hardness.

# Introduction

Type 2 diabetes mellitus (T2DM) affects bone homeostasis and quality, resulting in a three-fold increased risk of fragility fracture depending on the skeletal site and glycemic burden [1–5]. The disease burden is expected to increase further in the near future [6]. Therefore, early identification of bone disease and timely implementation of specific therapies can help prevent incident fragility fractures and subsequently reduce the substantial socioeconomic burden, morbidity and mortality.

At present, the measurement of areal bone mineral density (aBMD) is the standard approach to estimate bone health. However, aBMD is often normal or even slightly elevated in patients with T2DM compared to age- and sex-matched controls, hence, aBMD alone under-estimates fracture risk in T2DM [7,8]. Thus, mineral quantification alone, as represented by aBMD, is not sufficient, and the study of bone matrix (collagen) characteristics is equally important to predict future fracture risk [2,9–14]. Besides, bone histomorphometry is the gold standard to assess the bone quantity and quality [15,16], however, trans-iliac bone biopsy is an invasive procedure, requires expertise, and may not always be feasible to perform in routine clinical practice. Additionally, bone biopsy involves the risk of infection or delayed healing, particularly in individuals with uncontrolled hyperglycemia. Therefore, there is a need to develop newer techniques that can better assist and improve the accuracy of assessment of bone health in diabetes mellitus.

Our previous study have shown that the microstructural and macromolecular characteristics of fingernails are degraded in patients with T2DM [17]. Accordingly, we have hypothesized that the fingernail plate has the potential to serve as a potential surrogate marker to predict bone damage in diabetes because of the following reason: (i) The major constituent of fingernail plate (Keratin, present in ±85%) is also prone to glycation [17–20]. (ii) The growth of the nail plate is relatively slow; hence the long-term effects of hyperglycemia can reflect on the fingernail material and compositional properties [17,21]. (iii) Rate of fingernail growth and the bone cycle is 90–120 days, making it an ideal tissue to study. (iv) Study on fingernails is painless, non-invasive, and economical because they neither requires specific storage nor do they demand the use of expensive reagents [17–20].

Thus, the present study aims to investigate the link between bone and fingernail material/compositional properties in patients with T2DM. Herein, we have measured the microstructural properties (high-resolution μCT), material properties (reduced modulus and hardness), and compositional properties (relative content of glycation, normalized area of amide I and amide II) of both bone and fingernail.

# Research design and methods

## Study participants

Consecutive postmenopausal women admitted with femoral neck fragility fractures in the Department of Orthopedics, Post Graduate Institute of Medical Education and Research (PGI-MER), Chandigarh, India, undergoing hemi/total hip arthroplasty between July 2016 and July 2019 were recruited. All participants involved in the study were from Northern India. Patients with a prior history of fracture, having onychomycosis, on anti-osteoporotic medications, glucocorticoid, thiazides, or calcium/vitamin D supplements (over the last 6 months) were excluded. All patients underwent assessment of aBMD of the contralateral femoral neck using dual-energy X-ray absorptiometry (HOLOGIC Discovery A QDR 4500, Hologic Inc., Bedford, MA, USA). The study protocol was approved by the Institutional Ethics Committee, PGIMER, Chandigarh (Approval Number PGI/IEC/2015/171). Prior written informed consent was

obtained from all the participants that their discarded bone, clipped fingernail and clinical data will be used for scientific work. After exclusion, a total of 50 patients with included of whom 25 had T2DM while the rest 25 did not have diabetes mellitus. Later, all experiments were performed in Indian Institute of Technology Ropar.

## Sample procurement and storage

Excised femoral heads were collected from patients undergoing hemi/total hip arthroplasty. The cylindrical trabecular bone cores were extracted from each femoral head along the direction of the principal trabeculae using a drilling machine attached with a diamond core bit. The bone cores were cleaned with water jet, wrapped in saline-soaked gauze (PBS 7.4 pH), transferred into sample bags, labeled, and subsequently stored at -20˚C [14]. Along with the femoral head, the fingernail plate samples were also collected from the distal part of the right-hand middle finger using a nail clipper. Collected fingernail samples were sectioned into 2–3 small pieces so that they can be subsequently utilized for different characterization techniques [17]. All experiments were conducted within two months of sample collection. Experimental studies were performed in the Indian Institute of Technology (IIT), Ropar.

## Assessment of bone quality parameters

**Microstructural parameters by μ-CT.** One bone core of each patient was air-dried and scanned along the cylindrical axis using a high-resolution μCT system (GE Sensing & Inspection Technologies GmbH-phoenix|x-ray using 10 μm voxel size, 45 keV tube voltage, 250 μA beam current, 250 sec integration time, 10 frames). Reconstruction of scanned images was performed using Phoenix software (phoenix/x-ray, GE Measurement & Control; Germany), and reconstructed images were imported in Scan-IP (Simpleware Ltd, UK) for the analysis of the following structural parameters: bone volume fraction (BV/TV), trabecular thickness (Tb.Th), and trabecular number (Tb.N) were calculated as per published protocol [14].

**Fingernail and bone material properties by nanoindentation.** Nanoindentation tests were performed on both bone and fingernail samples using a TI-950 Tribo Indenter (Hysitron Inc., Minneapolis, MN, USA) with Berkovich pyramidal tip (tip radius ~150 nm). Before testing, the samples were embedded in epoxy and polished. Locations for indents were identified using an *in-situ* scanning probe microscope imaging integrated with a nanoindentation system, and all tests were performed at room temperature in a moist condition. Experiments were conducted as per the previously published protocol [14,17].

Eight indents with a peak load of 1000 μN were applied on the surface of the samples. A load function consisting of a ten-second ramp to peak force segment was adopted, followed by a thirty-second hold and a ten-second unloading segment [17]. The load-displacement curves obtained in these indentation tests were analyzed to determine the reduced modulus ($E_r$) and hardness ($H$) using Oliver and Pharr method in Triboscan (Hysitron) [22,23].

**Collagen and keratin properties by Fourier Transform Infrared Spectroscopy (FTIR).** The trabecular bone was demineralized using a 9.5% ethylene diamine tetra-acetic acid (EDTA) solution in phosphate-buffered saline. The tissue was submerged in the EDTA solution for five days at 4˚C; the solution was changed every 24 hours. After five days, the demineralized tissue was rinsed twice with acetone for 10 minutes and then rinsed twice with deionized water for 10 minutes [24]. Then, FTIR spectra were recorded from the freeze-dried fingernail and demineralized bone section using FTIR Spectrometer in Attenuated Total Reflectance (ATR) mode (Nicolet iS50, Thermo Scientific, Inc. Waltham, MA, USA) under constant pressure, in the spectral region of 4000 to 400 cm$^{-1}$. One sample of both bone and fingernail is tested of each donor with 4 μm resolution, and 60 scans were averaged. The

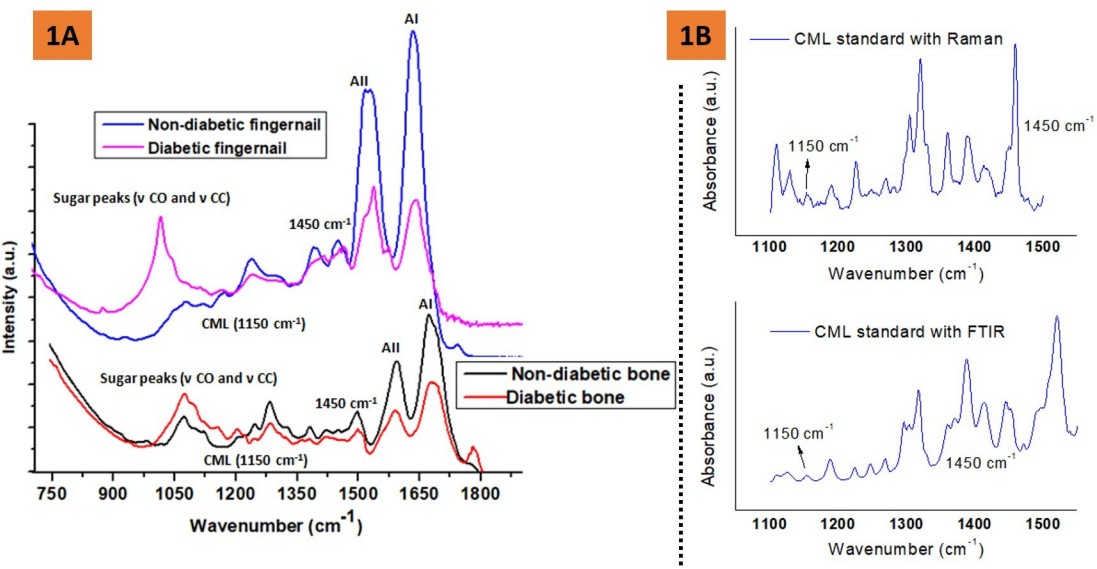

**Fig 1. A** Representative FTIR spectra of the human fingernail and trabecular bone (demineralized) showing the position of amide I, Amide II, CML, sugar, and methylene (CH₂) deformation band vibrations for the non-diabetic and diabetic groups, **B** showing Raman and FTIR spectra of a CML reference standard.

representative FTIR spectra of bone and fingernail with the appropriate label of various bands are shown in **Fig 1A.** The following parameters were calculated: the mean integrated area ratio (relative content) of amide I (protein C = O stretching, 1600–1700 cm−1) and Amide II (protein N–H bending, C–N stretching, 1500–1600 cm−1) bands with respect to methylene (CH₂) deformation band at 1450 cm$^{-1}$ [14,17,25,26].

The mean integrated area ratio (relative content) of sugar-to-matrix ratio [area of the sugar peak [(ν CO and ν CC peaks) (900–1100 cm$^{-1}$) to amide I peak (1596–1712 cm$^{-1}$)] was calculated as per published protocol [24].

The mean integrated area ratio (relative content) of 1150 cm$^{-1}$ (carboxymethyl-lysine, CML)/1450 cm$^{-1}$ was also calculated. Relative CML content has previously been validated with Raman spectroscopy [27,28], where CML is a lysine derivative and lysine reference has a significant Raman band at ~1150 cm$^{-1}$ [27,29]. Here, we have validated the presence of CML peak by both Raman and FTIR spectroscopies. For reference, both Raman and FTIR spectra of a CML standard (Cayman chemical company, Michigan, USA) have been included in **Fig 1B**.

**Statistical analysis.** The normality of the data was checked using the Kolmogorov-Smirnov test. Data were expressed as mean and standard deviation (SD) until otherwise specified. The homogeneity of variances was analyzed using the Levene's test. Between-group differences of calculated parameters were analyzed for statistical significance using Student's $t$-tests after testing for normality and homogeneity of variances. Pearson correlation and linear regression tests were used to determine relationships between structural, material, and compositional parameters. Observed power is calculated by comparing the mean value of sugar-to-matrix ratio in bone/fingernail between diabetic and non-diabetic groups using an ANOVA test. A confidence level of $p < 0.05$ implies a statistical significance between the groups where $p<0.05$, $p<0.01$, and $p<0.001$ denote the significance level.

Statistical analysis was performed using the Statistical Package for the Social Sciences (v.21, SPSS Inc., Chicago, IL, USA) and the Microsoft Office Excel (2007).

**Table 1. Baseline demographic and radiographic parameters of women with and without T2DM included in the present study.**

| Parameters | Women without T2DM (N = 25) | Women with T2DM (N = 25) | p-value |
|---|---|---|---|
| Age (years) | 74.1 ± 7.8 | 72.2 ± 8.7 | 0.701 |
| HbA1c (%) | 5.5 ± 0.6 | 7.8 ± 1.3 | **0.001** |
| Duration of diabetes (years) | na | 7.0 ± 2.2 | na |
| Femoral Neck aBMD (gm/cm$^2$) | 0.549 ± 0.079 | 0.487 ± 0.119 | 0.353 |
| Femoral Neck T score | -2.8 ± 0.87 | -2.9 ± 0.83 | 0.749 |

All data are expressed as mean ± SD, na: not applicable; HbA1c: glycosylated hemoglobin A1c; FN: Femoral neck; aBMD: areal bone mineral density.

* p<0.05, ** p<0.01 and ***p<0.001 respectively compared to non-diabetic group.

## Results

### Patient's characteristics

Over the study period, a total of 86 postmenopausal women with femoral neck fragility fractures were admitted in the Department of Orthopedics and underwent hemi/total hip arthroplasty. Out of these 86 women, 36 patients were excluded (18 had a recent history of calcium/vitamin D intake, 10 had a history of intake of anti-osteoporotic medications along with a recent history of calcium/vitamin D use, 5 had a prior history of fragility fractures, 3 did not provide written informed consent). Finally, 50 patients were included in the study. Out of these 50 patients, 25 patients had T2DM (mean age 74.1±7.8 years) and the rest 25 patients did not have T2DM (mean age 72.2±8.7 years). The mean duration of T2DM was 7.0±2.2 years; the mean HbA$_{1c}$ was 7.8±1.3%. There was no significant difference in age (p = 0.701) or contralateral femoral neck aBMD (p = 0.353) between the two groups (**Table 1**).

### Structural parameters (μ-CT)

Representative μCT images and mean values of microstructural parameters of diabetic and non-diabetic groups are shown in **Fig 2**. The diabetic bone had significantly lower (mean) values of BV/TV (17.0±4.4 vs. 22.1±6.1, **p = 0.009**), Tb.Th (0.146±0.03 vs. 0.166±0.03, **p = 0.028**)

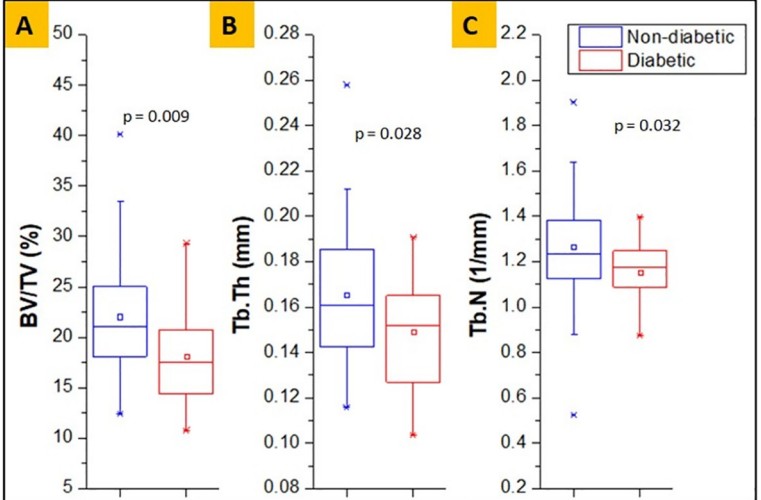

**Fig 2.** Measures of bone microstructural parameters for diabetic and non-diabetic trabecular bone, showing lower BV/TV (%) (A), Tb.Tb (mm) (B) and Tb.N (1/mm) (C) in diabetic group.

and Tb.N (1.15±0.13 vs. 1.26±0.25, **p = 0.032**) than the non-diabetic group. The diabetic group had significantly lower BV/TV (23.07%), Tb.Th (12.0%), and Tb.N (8.73%) compared to the non-diabetic group.

## Collagen and keratin properties and measurement of glycation

The bone of diabetic individuals had significantly higher sugar/matrix ratio (0.33±0.07 vs. 0.25±0.04, **p<0.001**) and relative content of CML (AGE) (0.74±0.32 vs. 0.54±0.27, **p = 0.019**) compared to non-diabetic individuals (**Fig 3A**). Likewise, the diabetic fingernail had significantly higher sugar/matrix ratio (0.30±0.10 vs. 0.18±0.07, **p<0.001**) and relative content of CML (AGE) (0.40±0.12 vs. 0.32±0.14, **p = 0.037**) compared to the non-diabetic group (**Fig 3B**).

Also, the diabetic bone had a lower value of area under the normalized peak of Amide I and Amide II bands (3.67±2.08 vs. 7.36±3.93, **p<0.001** and 1.22±0.91 vs. 2.93±1.49, **p<0.001**, respectively), compared to the non-diabetic group (**Fig 4A**). These results indicate that the quantity of these proteins is lower in the diabetic bone. Similarly, **Fig 4B** shows that the diabetic fingernail also had a lower value of area under the normalized peak of Amide I and Amide II bands (10.50±2.39 vs. 15.33±5.88, **p = 0.009** and 8.09±1.76 vs. 10.90±3.06, **p = 0.026**, respectively), compared to the non-diabetic group. These results also indicate that the quantity of these proteins is lower in the diabetic fingernail.

## Material properties

Nanoindentation tests for both the groups revealed that under the same load of 1000 µN, the diabetic bone had significantly lower (mean) values of reduced modulus (7.38±2.96 GPa vs. 9.13±2.58 GPa, **p = 0.022**) and hardness (0.268±0.16 GPa vs. 0.441±0.25 GPa, **p = 0.004**) than the non-diabetic group (**Fig 5A**). The modulus and hardness were lower by 19.17% and 39.23%, respectively, in the diabetic group compared to the non-diabetic group.

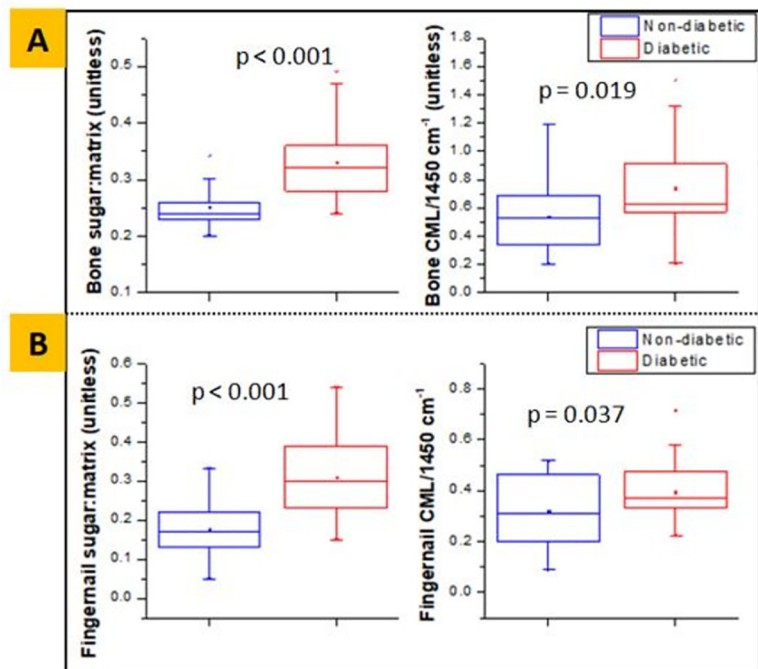

**Fig 3.** Sugar-to-matrix ratio and normalized CML content obtained from FTIR, showing higher mean values in diabetic group for both bone (A) and fingernail (B) respectively.

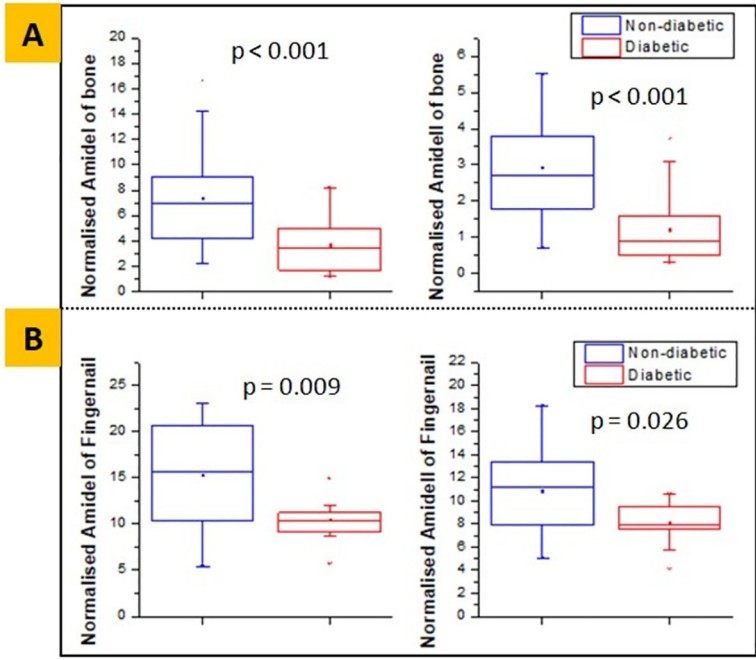

**Fig 4.** Normalized Amide I and Amide II area obtained from FTIR, showing lower mean values in diabetic group for both bone (**A**) and fingernail (**B**) respectively.

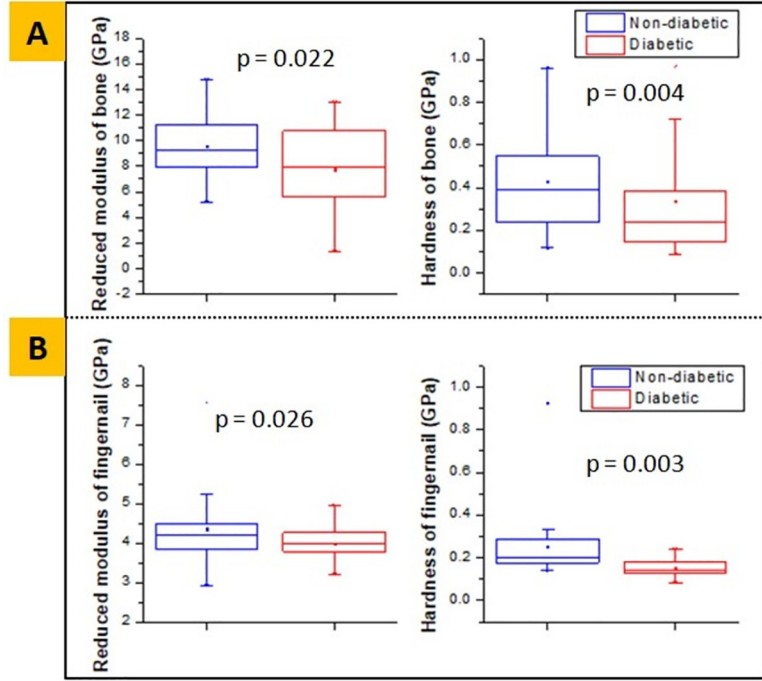

**Fig 5.** Reduced modulus and hardness obtained from nanoindentation, showing lower mean values in diabetic group for both bone (A) and fingernail (B) respectively.

With the same loading condition, the diabetic fingernail also had significantly lower (mean) values of reduced modulus (4.00±0.56 GPa to 4.60±0.98 GPa, **p = 0.026**) and hardness (0.152±0.04 GPa to 0.212±0.06 GPa, **p = 0.003**) than the non-diabetic group (**Fig 5B**). The reduced modulus and hardness were lower by 13.04% and 28.3%, respectively, in the diabetic fingernails compared to the non-diabetic subjects.

## Interrelationships between variables

In the diabetic group, the pre-operative HbA$_{1c}$ was positively correlated with the sugar-to-matrix ratio in bone (r = 0.654, **p<0.001**) and fingernail (r = 0.527, **p = 0.007**), as shown in **Fig 6A and 6B**, respectively.

Linear regression analyses with bone glycation as a dependent variable and fingernail glycation as an independent variable showed that in the diabetic group, the fingernail sugar-to-matrix ratio can explain up to 20% (r = 0.45, p = 0.025, β = 0.291) of variance in the sugar-to-matrix ratio in bone, and relative CML content of fingernail can explain up to 29% (r = 0.54, p = 0.005, β = 1.494) of variance in relative CML content in bone (**Fig 7A and 7B,** respectively). Also, the variance in reduced modulus of bone and microstructural parameter BV/TV can be explained by up to 23% (r = 0.48, p = 0.015, β = 2.937) and 21% (r = 0.46, p = 0.020, β = 5.477) (**Fig 7C and 7D),** respectively by the reduced modulus of fingernail in the diabetic group. In the non-diabetic group, none of the parameters were found to be significant.

We performed the observed power of the study by comparing the mean value of the sugar-to-matrix ratio in fingernail/bone between diabetic and non-diabetic groups, and this outcome was found to be 99.9% for both tissues.

## Discussion

The present study, which, to the best of our knowledge, happens to be the first of its kind, shows that the microstructural and material properties are altered in the bone of patients with T2DM, which go hand-in-hand with the fingernail plate compositional/material properties. We found significant correlations between the two body tissues, possibly suggesting that fingernails properties might serve as a potential surrogate marker of underlying bone quality in patients with T2DM.

In a prior study on an animal model, through the concurrent assessment of mechanical, microstructural, and compositional properties of bone, we had reported that T2DM alters the

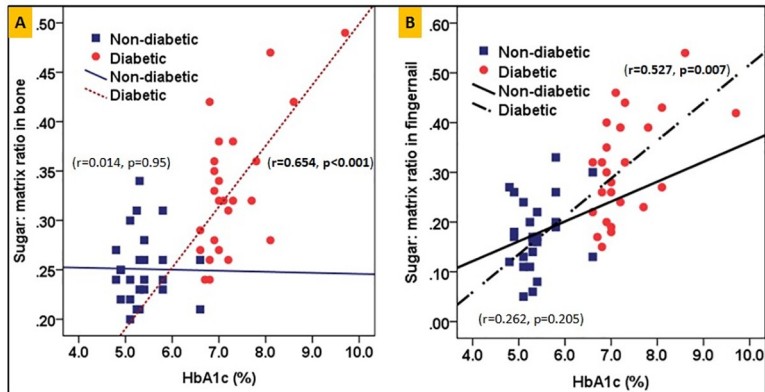

**Fig 6.** Correlation between HbA1c and the sugar-to-matrix ratio for both bone and fingernail are shown in 6(A) and 6 (B) respectively for diabetic and non-diabetics groups.

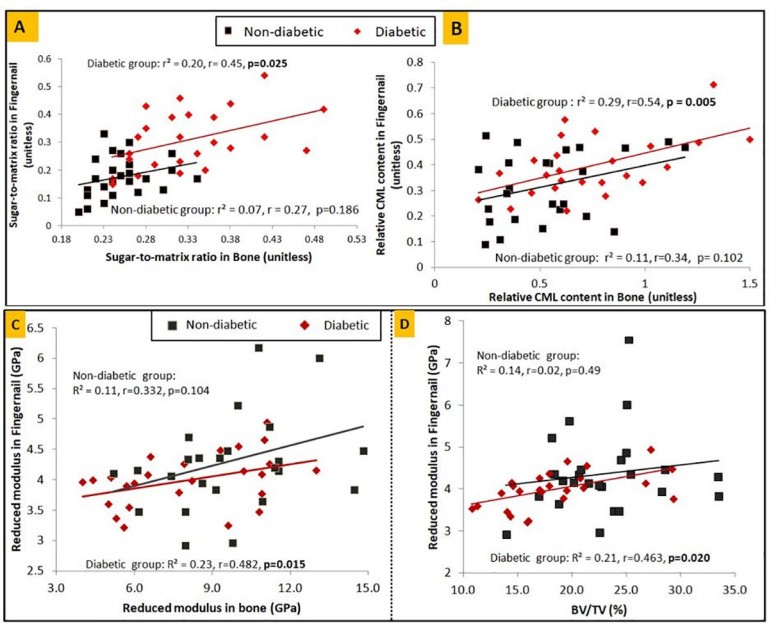

**Fig 7.** Correlation between fingernail and bone for diabetic patients and non-diabetics **(A)** for sugar-to-matrix ratio, **(B)** for relative CML content, **(C)** for nanoindentation determined reduced modulus, **(D)** BV/TV of trabecular bone and fingernail reduced modulus.

cortical bone quality making the bone weaker and prone to fragility fractures. The presence of T2DM reduces whole bone strength, compromises structural properties (μCT), and increases AGEs [30]. In another study on individuals with T2DM and known fragility fracture, we have shown that diabetes adversely affects the trabecular bone quality at multiple organization levels. The accumulation of AGEs favors the deterioration of bone quality in diabetes, leading to altered structural, material, and compositional properties. All these changes culminate in reduced bone elasticity and toughness with a corresponding increase in the propensity to bone fragility [14].

On the other hand, T2DM also exerts an adverse effect on the human fingernail plate quality. Hence, there could be an association in the degradation pattern of both the tissues. Indeed, the fingernail is composed predominantly of keratin which is prone to glycation just as collagen in bone. In addition, the fingernail's growth is slow, allowing time for hyperglycemia to exert its detrimental effect [17]. Carrying forward this hypothesis, in the present study, we found marked differences in compositional/material properties of bone and fingernail in patients with and without T2DM.

Here, the assessment of bone microstructure on micro-CT showed lower BV/TV (%) in those with diabetic group compared to the non-diabetic group. The bone microstructure was noticeably altered, evidenced by the thinning of trabeculae and by fewer trabecular numbers. However, few recent studies [24,31–33] obtained the same or increased BV/TV in those with diabetes compared to non-diabetics. Notably, in these studies [24,31–33], the bone tissues were obtained from individuals with obesity and/or severe arthritis, which possibly could explain the difference in the findings of BV/TV from this study. In contrast, the previous study published from our institution reported lower BV/TV in individuals with T2DM and fragility fractures [14]. It is also possible that our study finding of lower BV/TV in diabetes is related to the distinct phenotypes of Asians [30,34–38].

At the material level, a lower value of reduced modulus and hardness is observed in those with diabetes than in the non-diabetic group. Thus, this finding describes that long-standing T2DM alters bone microstructure and material properties and makes the bone weaker and fracture prone. These results also revealed that the above alteration in the properties locally, either at micro-scale or nano-scale, can affect the properties of the hierarchical organization of bone at higher scales [39,40].

In compositional analysis, we observed that the FTIR signatures and spectral changes of bone collagen and nail keratin are comparable and can discriminate the samples of the diabetic and non-diabetic groups. We found significantly increased glycation content (relative CML content and sugar-to-matrix ratio) in the diabetic group of both bone and fingernail compared to the non-diabetic group. Along with the increased glycation content, the decreased relative proteins content (Amide I/1450 cm$^{-1}$ and Amide II/1450 cm$^{-1}$) is observed in the diabetic groups for both bone and fingernail compared to the non-diabetic group. These results are consistent with the previous studies that reported the decreased total protein content in diabetic rat skeletal soleus muscles [25] and reduced Amide content in diabetic fingernails [17].

Furthermore, the HbA$_{1c}$ was significantly and positively correlated with the sugar-to-matrix ratio for both bone and fingernail, implying that the primary reason for the degradation of bone and fingernail quality in T2DM is prolonged hyperglycemia. Hyperglycemia, in turn, predisposes to the accumulation of AGEs in the bone and nail matrix and induces tissue damage through structural modification of proteins and abnormal proteins collagen fibril organization in diabetes [41]. Also, abnormal glycation decreases the osteogenic differentiation and activity [42,43], which results in reduced overall protein synthesis (decreased Amide I, Amide II content) and decreased material properties of both bone and fingernail at nano-scale, and lower bone formation (BV/TV) at micro-scale.

Usually, severe complications such as fracture or tearing are not observed in the diabetic nail. This is because the accumulation of AGE varies in the fingernail as it grows and gets entirely replaced within a few months, whereas the complete removal of accumulated AGEs from diabetic bone is not possible, and thus it causes severe damage and leads to bone fracture [17]. The present study revealed that in postmenopausal women with T2DM, the material and compositional properties of fingernails are degraded to a similar extent as is seen in the bones. The marked correlation observed for measures of glycation between fingernail and bone in the diabetic group confirms the role of fingernail in the prediction of bone glycation. Also, the reduced modulus of the fingernail was found to correlate with the reduced modulus of bone and BV/TV in the diabetic group, suggesting that the degraded fingernail material property is associated with the degraded bone material property and altered bone microstructure. Therefore, investigating fingernail properties can be a non-invasive solution to gauge about the status of bone health and the prediction of fracture risk in patients with diabetes mellitus.

We humbly acknowledge the study limitations. First, the sample size was limited to only 50 participants. Nevertheless, the index study was meant to be a pilot study. Second, we had included only postmenopausal women so as to ensure a homogenous study population. Inclusion of men and premenopausal women would have created multiple heterogeneous subgroups which would have compromised our statistical validity. Third, the collection of bone samples from the femoral head rather than the conventional trans-iliac bone biopsy. However, there exists a weak association between the histomorphometric parameters of the iliac crest and proximal femur. Hence, femoral head samples may be clinically more relevant in hip fractures [44].

In conclusion, the present pilot study shows that the microstructural and material properties are altered in the bone of patients with T2DM, which go hand-in-hand with the fingernail plate compositional/material properties. Accordingly, fingernails properties *might serve* as a

potential surrogate non-invasive marker of underlying bone quality in patients with T2DM. Early detection of impaired bone health in patients with T2DM would help timely implement bone-directed therapies that would avert incident fragility fractures. However, the findings need to be validated in further large-scale studies involving a substantial number of patients of both sexes with varying duration of diabetes and glycemic control.

## Acknowledgments

The IIT Ropar and PGIMER Chandigarh are highly acknowledged for providing the necessary infrastructure and facilities used in the current research.

## Author Contributions

**Conceptualization:** Praveer Sihota, Vijay G. Goni, Sanjay Kumar Bhadada.

**Data curation:** Deepak Neradi, Shailesh Karn, Siddhartha Sharma.

**Formal analysis:** Praveer Sihota, Rimesh Pal, Ram Naresh Yadav.

**Funding acquisition:** Navin Kumar.

**Investigation:** Praveer Sihota, Ram Naresh Yadav.

**Methodology:** Praveer Sihota.

**Project administration:** Navin Kumar.

**Resources:** Vijay G. Goni, Siddhartha Sharma, Vishwajeet Mehandia, Navin Kumar.

**Supervision:** Vishwajeet Mehandia, Sanjay Kumar Bhadada, Navin Kumar.

**Validation:** Praveer Sihota, Sanjay Kumar Bhadada.

**Visualization:** Sanjay Kumar Bhadada.

**Writing – original draft:** Praveer Sihota.

**Writing – review & editing:** Praveer Sihota, Rimesh Pal, Vijay G. Goni, Vishwajeet Mehandia, Sanjay Kumar Bhadada, Navin Kumar, Sudhaker D. Rao.

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
