## [Decision Letter · Decision Letter 0]

10 Aug 2021

PONE-D-21-04313

Can Fingernail Quality Predict the Bone Damage in Type 2 Diabetes Mellitus: A Pilot Study

PLOS ONE

Dear Dr. KUMAR,

Thank you for submitting your manuscript to PLOS ONE. After careful consideration, we feel that it has merit but does not fully meet PLOS ONE’s publication criteria as it currently stands. Therefore, we invite you to submit a revised version of the manuscript that addresses the points raised during the review process.

Specifically, the authors should address the issues that relate to: 1) softening of the study conclusion based on the small sample size; 2) limitations of the study; and 3) grammatical errors.

We look forward to receiving your revised manuscript.

Kind regards,

Subburaman Mohan

Academic Editor

PLOS ONE

Journal Requirements:

3. In your Methods section, please provide additional information about the participant recruitment method and the demographic details of your participants. Please ensure you have provided sufficient details to replicate the analyses such as: a) the recruitment date range (month and year),b) descriptions of where participants were recruited and where the research took place

4. Please ensure you have discussed the rationale for limiting the study to female patients, and ensure that your conclusions are presented appropriately given this potential limitation.

5. Please provide a sample size and power calculation in the Methods, or discuss the reasons for not performing one before study initiation.

Reviewers' comments:

Reviewer's Responses to Questions

**Comments to the Author**

1. Is the manuscript technically sound, and do the data support the conclusions?

Reviewer #1: Partly

Reviewer #2: Yes

2. Has the statistical analysis been performed appropriately and rigorously? 

Reviewer #1: Yes

Reviewer #2: I Don't Know

3. Have the authors made all data underlying the findings in their manuscript fully available?

Reviewer #1: Yes

Reviewer #2: Yes

4. Is the manuscript presented in an intelligible fashion and written in standard English?

Reviewer #1: Yes

Reviewer #2: Yes

5. Review Comments to the Author

Reviewer #1: Overall, interesting article and adds to the pool of diabetes and bone research

Some areas that require grammatical changes include:

1." This finding evidence that the main reason for degradation of bone and fingernail quality in the diabetic group is prolonged....".

2. "present study revealed that the material and compositional properties of the fingernail is degrading as similar to the bone in individuals with diabetes.."

The conclusion is somewhat strong given that the fingernail glycation only explained roughly 30% of the the bone glycation changes in the very small sample size.

Reviewer #2: Diabetes mellitus with the effect on bone cells has deleterious effects on bone metabolism and bone quality. It seems that hyperglycemia leads to accelerated bone loss, osteopenia and osteoporosis.

In diabetic patients, including Type 2 Diabetes Mellitus (T2DM), Bone Mineral Density (BMD) only assesses bone strength. So other techniques are needed to assess bone tissue quality in T2DM patients. Your study was designed for this purpose.

In this study, the link between material/compositional properties of bone and fingernails were investigated.

Femoral head and fingernail samples were obtained from twenty-five adult female patients (with/without T2DM) with fragility femoral neck fractures undergoing hemi/total hip arthroplasty. Bone volume fraction was lower in diabetic group than nondiabetic group due to fewer and thinner trabeculae. Both bone/fingernail in T2DM had lower reduced modulus, hardness, lower Amide I and Amide II area ratio (protein content), higher sugar-to-matrix ratio and relatively high carboxymethyl-lysine (CML) content compared with non-diabetic patients. Sugar-to-matrix ratio and relative CML content were strongly and positively correlated with HbA1c for both bone/fingernail.

In this study there was a positive correlation between bone and fingernail glycation content. Degradation pattern of bone and fingernail properties go hand-in-hand in individuals with T2DM. Based on this study fingernail compositional/material properties can serve as a noninvasive surrogate marker of bone quality in T2DM.

One of the limitations of this study is the small number of patients (small sample population). Another limitation is that the all patients were women. It is suggested that other studies are designed with more patients (larger sample size), both genders (male and female), diabetic patients from a younger age range, and patients with type 1 diabetes.

6. PLOS authors have the option to publish the peer review history of their article (what does this mean?). If published, this will include your full peer review and any attached files.

Reviewer #1: No

Reviewer #2: No

---

## [Author Response · Author response to Decision Letter 0]

11 Sep 2021

We thank the Reviewers for their time to review our manuscript and appreciate their suggestions. We have addressed the Reviewers' and Editor comments and incorporated corresponding changes in the revised manuscript (highlighted in yellow). 

Below we provide specific responses to the Reviewers' and Editor's comments/suggestions:

Reply to comments from Reviewer-1

Comments: Overall, interesting article and adds to the pool of diabetes and bone research.

Comment 1 

Some areas that require grammatical changes include:

1." This finding evidence that the main reason for degradation of bone and fingernail quality in the diabetic group is prolonged....".

2. "Present study revealed that the material and compositional properties of the fingernail is degrading as similar to the bone in individuals with diabetes."

Response: Grammatical error is corrected in the above sentences in the revised manuscript. Also, we have rechecked the entire manuscript and minimized the grammatical error as much as possible. 

Comment 2 The conclusion is somewhat strong given that the fingernail glycation only explained roughly 30% of the bone glycation changes in the very small sample size.

Response: The conclusion is edited as per the suggestion, and the revised conclusion is as below:

In conclusion, the present pilot study shows that the microstructural and material properties are altered in the bone of patients with T2DM, which go hand-in-hand with the fingernail plate compositional/material properties. Accordingly, fingernails properties might serve as a potential surrogate non-invasive marker of underlying bone quality in patients with T2DM. Early detection of impaired bone health in patients with T2DM would help timely implement bone-directed therapies that would avert incident fragility fractures. However, the findings need to be validated in further large-scale studies involving a substantial number of patients of both sexes with varying duration of diabetes and glycemic control.

Response to comments from Reviewer-2

Comments: Diabetes mellitus with the effect on bone cells has deleterious effects on bone metabolism and bone quality. It seems that hyperglycemia leads to accelerated bone loss, osteopenia and osteoporosis. In diabetic patients, including Type 2 Diabetes Mellitus (T2DM), Bone Mineral Density (BMD) only assesses bone strength. So other techniques are needed to assess bone tissue quality in T2DM patients. Your study was designed for this purpose. In this study, the link between material/compositional properties of bone and fingernails were investigated. Femoral head and fingernail samples were obtained from twenty-five adult female patients (with/without T2DM) with fragility femoral neck fractures undergoing hemi/total hip arthroplasty. Bone volume fraction was lower in diabetic group than nondiabetic group due to fewer and thinner trabeculae. Both bone/fingernail in T2DM had lower reduced modulus, hardness, lower Amide I and Amide II area ratio (protein content), higher sugar-to-matrix ratio and relatively high carboxymethyl-lysine (CML) content compared with non-diabetic patients. Sugar-to-matrix ratio and relative CML content were strongly and positively correlated with HbA1c for both bone/fingernail. In this study there was a positive correlation between bone and fingernail glycation content. Degradation pattern of bone and fingernail properties go hand-in-hand in individuals with T2DM. Based on this study fingernail compositional/material properties can serve as a noninvasive surrogate marker of bone quality in T2DM.One of the limitations of this study is the small number of patients (small sample population). Another limitation is that the all patients were women. 

It is suggested that other studies are designed with more patients (larger sample size), both genders (male and female), diabetic patients from a younger age range, and patients with type 1 diabetes. 

Response: Thank you for the detailed summary of our work and suggestions. 

Reply to comments from Editor

Comment 1: Specifically, the authors should address the issues that relate to: 1) softening of the study conclusion based on the small sample size; 2) limitations of the study; and 3) grammatical errors.

Response: 

Manuscript has been revised according to all three suggestions. 

1. The conclusion is edited in the revised manuscript according to suggestions. 

In conclusion, the present pilot study shows that the microstructural and material properties are altered in the bone of patients with T2DM, which go hand-in-hand with the fingernail plate compositional/material properties. Accordingly, fingernails properties might serve as a potential surrogate non-invasive marker of underlying bone quality in patients with T2DM. Early detection of impaired bone health in patients with T2DM would help timely implement bone-directed therapies that would avert incident fragility fractures. However, the findings need to be validated in further large-scale studies involving a substantial number of patients of both sexes with varying duration of diabetes and glycemic control.

2. Limitation of the study is incorporated in revised manuscript

We humbly acknowledge the study limitations. First, the sample size was limited to only 50 participants. Nevertheless, the index study was meant to be a pilot study. Second, we had included only postmenopausal women so as to ensure a homogenous study population. The inclusion of men and premenopausal women would have created multiple heterogeneous subgroups which would have compromised our statistical validity. Third, the collection of bone samples from the femoral head rather than the conventional trans-iliac bone biopsy. However, there exists a weak association between the histomorphometric parameters of the iliac crest and proximal femur. Hence, femoral head samples may be clinically more relevant in hip fractures [44]. 

3. Grammatical errors are minimized as per best of our knowledge

Additional Journal Requirements:

Comment 1: Please ensure that your manuscript meets PLOS ONE's style requirements, including those for file naming.

Response: Corrected as per the suggestion

Comment 2: Please review your reference list to ensure that it is complete and correct.

Response: The reference list is edited as per Plos One style, and it is rechecked. 

Comment 3: In your Methods section, please provide additional information about the participant recruitment method and the demographic details of your participants. Please ensure you have provided sufficient details to replicate the analyses such as: a) the recruitment date range (month and year), b) descriptions of where participants were recruited and where the research took place

Response: Revised as below:

Consecutive postmenopausal women admitted with femoral neck fragility fractures in the Department of Orthopedics, Post Graduate Institute of Medical Education and Research (PGIMER), Chandigarh, India, undergoing hemi/total hip arthroplasty between July 2016 and July 2019 were recruited. All participants involved in the study were from Northern India. Patients with a prior history of fracture, having onychomycosis, on anti-osteoporotic medications, glucocorticoid, thiazides, or calcium/vitamin D supplements (over the last 6 months) were excluded. All patients underwent assessment of aBMD of the contralateral femoral neck using dual-energy X-ray absorptiometry (HOLOGIC Discovery A QDR 4500, Hologic Inc., Bedford, MA, USA). The study protocol was approved by the Institutional Ethics Committee, PGIMER, Chandigarh (Approval Number PGI/IEC/2015/171). Prior written informed consent was obtained from all the participants that their discarded bone, clipped fingernail and clinical data will be used for scientific work. After exclusion, a total of 50 patients with included of whom 25 had T2DM while the rest 25 did not have diabetes mellitus. Later, all experiments were performed in Indian Institute of Technology Ropar. 

Also in result section:

Over the study period, a total of 86 postmenopausal women with femoral neck fragility fractures were admitted in the Department of Orthopedics and underwent hemi/total hip arthroplasty. Out of these 86 women, 36 patients were excluded (18 had a recent history of calcium/vitamin D intake, 10 had a history of intake of anti-osteoporotic medications along with a recent history of calcium/vitamin D use, 5 had a prior history of fragility fractures, 3 did not provide written informed consent). Finally, 50 patients were included in the study.

Comment 4: Please ensure you have discussed the rationale for limiting the study to female patients, and ensure that your conclusions are presented appropriately given this potential limitation

Response: Suggested limitation is included in the revised manuscript discussion section. 

Second, we had included only postmenopausal women to ensure a homogenous study population. The inclusion of men and premenopausal women would have created multiple heterogeneous subgroups which would have compromised our statistical validity.

Comment 5: Please provide a sample size and power calculation in the Methods, or discuss the reasons for not performing one before study initiation

Response: Sample size is added in the Methods section, n=25 in each group. Also, observed power is calculated using an ANOVA test and reported in the revised manuscript. 

Observed power is calculated in the revised manuscript using SPSS (v.21, SPSS Inc., Chicago, IL, USA) by comparing the mean value of the sugar-to-matrix ratio in fingernail/bone between diabetic and non-diabetic groups using an ANOVA test, with an alpha of 0.05. The power of the study for this outcome was 99.9% for both tissues.

Once again, we thank the reviewers and the Editor for their constructive comments. The manuscript is much improved, and we hope it is now acceptable for publication in the journal. 

Sincerely

Dr. Navin Kumar

---

## [Editor Report · Decision Letter 1]

15 Sep 2021

Can Fingernail Quality Predict the Bone Damage in Type 2 Diabetes Mellitus?: A Pilot Study

PONE-D-21-04313R1

Dear Dr. KUMAR,

We’re pleased to inform you that your manuscript has been judged scientifically suitable for publication and will be formally accepted for publication once it meets all outstanding technical requirements.

Kind regards,

Subburaman Mohan

Academic Editor

PLOS ONE

---

## [Editor Report · Acceptance letter]

23 Sep 2021

PONE-D-21-04313R1 

Can Fingernail Quality Predict Bone Damage in Type 2 Diabetes Mellitus? A Pilot Study 

Dear Dr. Kumar:

I'm pleased to inform you that your manuscript has been deemed suitable for publication in PLOS ONE. Congratulations! Your manuscript is now with our production department. 

Kind regards, 

on behalf of

Dr. Subburaman Mohan 

Academic Editor

PLOS ONE